# The Potentiality of Plant-Derived Nanovesicles in Human Health—A Comparison with Human Exosomes and Artificial Nanoparticles

**DOI:** 10.3390/ijms23094919

**Published:** 2022-04-28

**Authors:** Mariantonia Logozzi, Rossella Di Raimo, Davide Mizzoni, Stefano Fais

**Affiliations:** 1Department of Oncology and Molecular Medicine, Istituto Superiore di Sanità, Viale Regina Elena 299, 00161 Rome, Italy; 2ExoLab Italia, Tecnopolo d’Abruzzo, Strada Statale 17 Loc. Boschetto di Pile, 67100 L’Aquila, Italy; rossella@exolabitalia.com (R.D.R.); davide@exolabitalia.com (D.M.)

**Keywords:** extracellular vesicles, nanovesicles, plants, therapy, drug delivery, organic agriculture, health

## Abstract

Research in science and medicine is witnessing a massive increases in literature concerning extracellular vesicles (EVs). From a morphological point of view, EVs include extracellular vesicles of a micro and nano sizes. However, this simplistic classification does not consider both the source of EVs, including the cells and the species from which Evs are obtained, and the microenvironmental condition during EV production. These two factors are of crucial importance for the potential use of Evs as therapeutic agents. In fact, the choice of the most suitable Evs for drug delivery remains an open debate, inasmuch as the use of Evs of human origin may have at least two major problems: (i) autologous Evs from a patient may deliver dangerous molecules; and (ii) the production of EVs is also limited to cell factory conditions for large-scale industrial use. Recent literature, while limited to only a few papers, when compared to the papers on the use of human EVs, suggests that plant-derived nanovesicles (PDNV) may represent a valuable tool for extensive use in health care.

## 1. Introduction

In the last few years, we have witnessed a dramatic growth of studies on extracellular vesicles (EVs). These vesicles, which can take part in different physiological mechanisms [1], have been gaining attention as potential therapeutics [2,3]. In this context, the presence of EVs in food is also being increasingly considered. A large number of studies have been published, related to mammalian cell-derived EVs, including products as food sources, though the existence of EVs in plants is still controversial.

The presence of exosomes (or endosomal-derived vesicles) in foods has been included in the FAO/INFOODS databases, and been named “FoodEVs”. In fact, the information about food EVs is limited to four FAO groups, namely, milk, starchy roots and tubers, nuts and seeds, and fruits; however, they have been proposed as ideal nanocarriers for future therapeutic strategies [4].

Previous studies have suggested that nanosized particles from plant cells may be exosome-like [5], and that nanoparticles derived from edible plants (grape, grapefruit, ginger, and carrots) show therapeutic properties against a variety of diseases [6,7,8,9,10], including inflammation [11,12,13,14,15,16,17], tumors [18,19,20,21,22,23], and infectious conditions [18,20,24,25,26,27].

Citrus fruits are the most economically important fruit crop in the world and constitute one of the main food-derived sources of vitamin C (Vit. C). Citrus fruits also contain the largest number of carotenoids found in any fruit, along with an extensive array of secondary compounds with pivotal nutritional properties. These substances have been shown to have anti-cancer, anti-oxidant, anti-inflammatory, anti-cholesterolemic, and anti-allergic activities [28]. Recent investigations showed for the first time that bioactives are released as plant-derived secretome in the apoplast of the juice sac, and are possibly also packed into exosome-like nanoparticles. As an example, exosome-like nanovesicles from *Citrus limon* L. (EXO-CLs) have shown anti-oxidant capacity and therapeutic activity with potential use against a variety of diseases [29,30].

Now, we have substantial information on the potential role of plant-derived nanovesicles in delivering molecules of a different nature [4,7,16,31,32,33,34,35,36,37,38,39,40,41,42,43,44,45,46,47,48]. A considerable number of nanoparticles can be purified from edible fruits. An important issue is the ability of fruit-derived nanovesicles to deliver, both in vitro and in vivo, a variety of therapeutic agents, including chemotherapeutic drugs, DNA expression vectors, siRNA, and new therapeutic molecules, such as antibodies. Importantly, fruit-derived nanovesicles (FDNVs) can be modified to achieve specific cellular targeting, in turn suggesting that these nanovesicles are excellent candidates to deliver an increasing variety of therapeutic agents [4]. FDNVs can be obtained from a broad family of citrus fruits, which include they *Citrus limon*, *Citrus paradisi*, *Citrus medica*, *Citrus reticulata*, *Citrus sinensis*, *Citrus bergamia*, *Citrus clementina*, *Citrus monstruosa*, and many other varieties that are diffused worldwide. Only in a few countries, including Italy, are the above-listed citrus fruits natural products obtainable from organic agriculture. In fact, we know that, generally, nanovesicles, including human exosomes, represent a way to eliminate extracellularly toxic molecules, such as chemical drugs [49,50] or theranostic nanoparticles [51]. Thus, where plant-derived nanovesicles are thought to be used as natural nanovectors, vegetables deriving from intensive agriculture should be avoided as FDNVs, due to the vehicle of elimination of toxic molecules (e.g., pesticides) or, more generally, unwanted material. This problem also involves OGM-technology-derived plants. It is mandatory that plants be cultured in an organic way, not being treated with pesticides, so as to ensure that their FDNVs are empty of toxic materials. This is a paramount advantage in exploiting plant-derived nanovesicles for any kind of therapeutic use, from supplementation, to dermo-cosmetics, to drug delivery. Starting from this point, plant-derived nanovesicles may well be used for further manipulation; for instance, to increase the level of targeting to a specific human body site. Moreover, it has been shown that FDNVs obtained from the same fruits differ both qualitatively and quantitatively when coming from either intensive or organic agriculture [30]. There is another key advantage of plant-derived nanovesicles; they can be obtained continuously and directly from fresh fruits and vegetables, without needing giant cell factories, such as those needed for human cell-derived exosomes. This advantage can dramatically reduce production costs.

Thus, we want to propose a review of the existing literature on the characterization and the use of plant-derived nanovesicles, with the purpose of improving the health of human beings. The readers will learn that plant-derived nanovesicles have many more advantages than disadvantages when compared to human exosomes, and that they offer us a chance to return to natural options.

## 2. Comparing Human Exosomes to Plant Nanovesicles

Human exosomes have been extensively studied in the last few decades, and without any doubt they represent a melting pot of new discoveries in science, and, above all else, new knowledge on the complex and integrated way our bodies works [1,2,3]. We know that after their release from an organ or compartment exosomes travel through the body. This probably represents a dynamic network, through which our organs and compartments are continuously crosstalking, with the aim of maintaining a dynamic homeostasis. Along with representing a source of proteins, lipids, and nucleic acids, they are considered a real and natural delivery system for disease biomarkers. Some recent reports have presented clinical data showing that these may change in number and size, depending on the disease state [52,53,54,55,56]. Human exosomes can be counted and characterized using NTA, immunocapture-based ELISA, or nanoscale flow cytometry [52,53,54,55,56]. In particular, immunocapture-based technologies may allow a large-scale and extended analysis, including both exosomal and cellular source markers, which provide the most complete exosome passport [57]. These approaches can show that exosomes may be detected in different body fluids and under different disease conditions, including cardiovascular injuries [58] and transmissible diseases [59,60]. All this information supports the use of human exosomes in setting new diagnostic approaches [2]; however, many concerns have been raised regarding the use of human exosomes for the delivery of therapeutic molecules. This, on the one hand, is due to the limited large-scale production of human exosomes from normal human cells. On the other hand, there is evidence that human exosomes may represent one of the scavenging mechanisms of our body [49,50,51], and, too often, they deliver potentially dangerous materials, including tumor derived molecules [61], foreign nuclei acids [62], and transmissible agents [59,60]. Thus, together with the ongoing use of artificial nanoparticles, the use of plant-derived extracellular vesicles is considered, as they appear safer while having comparable structures and functions, including that of a natural interspecies delivery system. In fact, there are many features that human exosomes have in common with vegetable- and fruit-derived nanovesicles (VFNVs), including size, charge, and similar lipid composition [43]; however, a major difference is that VFNVs contain bioactives that are key for our metabolism, and, only very recently, has it been shown that the molecules contained in VFNVs exhibit clear enzymatic activity [30]. This is a very important point, for instance, for free Vit. C as it is currently contained in commercially available products, has some limits due to its low level of bioavailability, in turn due to the high level of instability of the molecule that is thermolabile and photosensitive, and needing oxidation to become absorbable by gut mucosa. Previous data have shown that VFNVs have a more potent anti-oxidant effect, probably due to their ability to protect anti-oxidants from both quick oxidation and degradative attack from gastric juice and bile. Now, we know the contents of ascorbic acid and other anti-oxidants in a broad range of fruit-derived nanovesicles [30]; however, when comparing VFEVs purified from fruits and vegetables coming from organic farming to those obtained from intensive farming, there were substantial differences in terms of both the amount of nanovesicles and the anti-oxidant capacity [30]. VFNVs from organic agriculture showed higher number of nanovesicles and greater anti-oxidant capacity, with a size in the range of nm. Thus, we should start thinking of a new acronym for vegetable and fruit nanovesicles (VFNVs). Further analysis has shown that VFNVs contain variable amounts of ascorbic acid, glutathione, superoxide dismutase-1 (SOD-1), and catalase. We know that these groups of bioactives are involved in many functions of our body, including collagen formation, iron absorption, various cellular metabolisms, and the immune response.

In summary, the advantages of nanovesicles from organic agriculture are the following: (i) they are natural nanovesicles offered by nature that can be used to provide a valuable tool for cell-to-cell communications; (ii) their lipid membranes protect bioactives from external agents, including pH and thermal and light variations; (iii) they undergo a natural mechanism of cellular uptake through membrane-to-membrane fusion, and, through this mechanism, they can pass some barriers, including the brain barrier and the placenta; (iv) they are tolerated by the immune system, inasmuch as they are contained in foods currently consumed by human beings; (v) they have a verified scalability and thus are suitable for industrial applications; and (vi) they are non-toxic, as they are derived from fruits and vegetables from organic agriculture. Thus, the first claim that we can propose is that nanovesicles extracted from organic agriculture-derived fruits and vegetables may represent “the ideal nanovector for any sort of therapeutic molecule”.

## 3. The Role of Plant-Derived Nanovesicles in Supplementation and Drug Delivery

### 3.1. The Role of Plant-Derived Nanovesicles in Future Approach of Supplementation

As mentioned before, plant-derived nanovesicles are a group of nano-scale vesicles that are isolated from dietary vegetables and fruits [10,43,63]. Existing nanovesicles and nanoparticles include a broad range of membrane-bounded structures from distinct origins, including (1) natural extracellular vesicles (including exosomes), as well as intracellular vesicles existing in plants; (2) artificial vesicles formed during preparation and extraction processes; and (3) synthetic liposomes or nanovectors prepared from plant-derived molecules as natural vesicle mimics. However, plant-derived nanovesicles are of paramount interest for their potential applications as natural suppliers. In terms of their structures, we know that they include a series of anti-oxidants (ascorbic acid, glutathione, superoxide dismutase-1, catalase), lipids, proteins, nucleic acids, and secondary metabolites (Figure 1). Most of all, they have a lipid-enriched membrane that contains a series of anti-oxidants that are entirely bioavailable, thus representing a valid alternative to the synthetic anti-oxidants that are currently available in pharmacies.

Moreover, plant-derived nanovesicles have, by definition, a high level of biocompatibility, being that they are contained in currently consumed foods, and the potential for large-scale production. More and more evidence suggests that plant-derived nanovesicles can easily enter mammalian cells and mediate plant–animal cross-kingdom gene regulation, i.e., plant small RNAs packed in plant-derived nanovesicles that could survive in an active form in animals and exogenously modulate host cellular processes via genetic crosstalk [63]. This indicates a potential medical application for plant-derived nanovesicles in the regulation of fundamental biological processes in the human body. Current literature suggests that plant-derived nanovesicles may have a role in: (i) homeostatic regulation of the immune system [8,64]; (ii) development of tissue engineering and reconstruction [65]; and (iii) delivery of therapeutic molecules of various origins [17,43,63,66,67] (Table 1).

For the above reasons, these nanovesicles can be used as nutraceutics, cosmetics, and regenerative products.

### 3.2. The Role of Plant-Derived Nanovesicles in Drug Delivery

Pharmaceutical research has been supporting innovation on the identification of “nanostructure” for at least three decades, which are suitable for ideal delivery of therapeutic molecules to disease sites, in order to increase the efficacy and reduce unwanted side effects [16]. The idea was to build lipid rafts that enclose therapeutic molecules in order to avoid unwanted degradation before arriving at the disease site. These nanoparticles are called “liposomes” and they are currently the only medical tool that have been tested in clinical studies. Liposomes have been designed to allow the encapsulation of both hydrophilic molecules, such as siRNA, RNA, and DNA (into the vesicle aqueous core), and hydrophobic bioactive compounds, such as proteins, phenolic peptides, and antibodies (into the lipid layer). Liposomes are obtained through several techniques, including membrane extrusion, sonication, micro-emulsification, and freezing–thawing cycling. These technical approaches have been used in liposome formulations with amphoterin B, doxorubicin, verteporfin, citarabin, morphine sulfate, and daunorubicin [68], and, more recently, for SARS-CoV-2 vaccines (Pfizer and Moderna).

However, liposome preparation may be problematic due to the many chemical treatments needed for modifying the lipid layers in order to include the therapeutic molecules [69,70]. Moreover, clinical use has shown a high level of toxicity due to the massive sequestration of liposomes into the filter organs (e.g., liver and spleen), thus increasing systemic toxicity, with a delusive level of efficiency.

On the other hand, some recent publications have shown the capacity of human exosomes to deliver molecules of a different nature, such as drugs [49,50] and theranostic nanoparticles [51]. Moreover, it has been shown that, as human nanovesicles are natural carriers for both bioactive molecules [71] and therapeutic antibodies [72], it seems conceivable that plant-derived nanovesicles may exert the same functions, due to the many analogies with their human counterparts. In fact, similar to human nanovesicles, plant-derived nanovesicles have also shown a natural ability to deliver chemical molecules [42].

Due to their lipid structure (very similar to the plasma membrane structure), plant-derived nanovesicles may protect their contents against external agents. For this reason, they are natural nanocarriers of bioactive compounds. An important added value is the high level of resistance of nanovesicle lipid membranes, which has been shown using sonication [72]. Some examples are (i) nanovesicles from ginger that have been used to deliver doxorubicin using electrostatic interactions [16]; and (ii) grapefruit nanovesicles charged with paclitaxel have been successfully administered intranasally in murine models [42]. Both these pieces of evidence provide a proof of concept, supporting the use of plant-derived nanovesicles as a new, very efficient and side-effect-free approach to nano-delivery of therapeutic molecules. There is an enthusiastic general interest in the use of plant-derived nanovesicles for drug delivery [44,45], which have been shown not to undergo filter organs sequestration, with limited or no systemic toxicity [73].

In addition, compared to the currently available drug delivery systems, plant-derived nanovesicles have multiple advantages, such as low immunogenicity and stability in the gastrointestinal tract [4,43].

**Table 1 ijms-23-04919-t001:** Application of plant-derived nanovesicles.

Source	Derivation	Application	Reference
**Aloe vera (*Aloe vera barbadensis*)**	Non-specified	Vesicles are efficiently taken up by bone marrow-derived macrophages and inhibit activation of NLRP3 inflammasome	[74]
**Apple (*Malus domestica*)**	Non-specified	Apple nanovesicles exert a potent anti-inflammatory effect in vitro	[64]
**Arabidopsis (*Arabidopsis thaliana*)**	Non-specified	EVs secretion is enhanced during biotic stress and EVs proteome changes in response to *P. syringae* infection	[32]
**Blueberry (*Vaccinium myrtillus, Vaccinium corymbosum*)**	Non-specified	Characterization of small RNAs in nanovesicles; miRNAs regulate the expression of inflammatory cytokines and cancer-related genes in vitro	[75]
**Broccoli (*Brassica oleracea*)**	Non-specified	Broccoli-derived nanoparticles inhibit colitis in treated mice	[8]
**Cactus (*Cactus*)**	Non-specified	Vesicles are efficiently taken up by bone marrow-derived macrophages and inhibit activation of NLRP3 inflammasome	[74]
**Carrot (*Daucus carota*)**	Non-specified	Isolated vesicles are taken up by intestinal macrophages; vesicles induce Nrf2 expression	[76]
	Non-specified	miRNAs of plant-derived nanovesicles influence microbiota composition	[77]
	Non-specified	Nanovesicles isolated from carrots exert potent anti-oxidative and apoptotic effects in in vitro cardiomyoblasts and neuroblastoma cell lines	[19]
**Cilantro (*Coriandrum sativum*)**	Non-specified	Vesicles are efficiently taken up by bone marrow-derived macrophages and inhibit activation of NLRP3 inflammasome	[74]
**Clementine (*Citrus clementina*)**	Non-specified	Characterization of membrane transporters in nanovesicles derived from clementine juice	[36]
**Coconut (*Cocos nucifera*)**	Non-specified	Characterization of small RNAs in nanovesicles; miRNAs regulate the expression of inflammatory cytokines and cancer-related genes in vitro	[75]
	Non-specified	Characterization of miRNAs in extracellular vesicles isolated from immature coconut water and mature coconut water	[78]
**Cucumber (*Cucumis sativus*)**	Non-specified	Characterization of cucumber-derived exosome-like vesicles	[79]
**Dandelion (*Taraxacum*)**	Non-specified	Vesicles are efficiently taken up by bone marrow-derived macrophages and inhibit activation of NLRP3 inflammasome	[74]
**Garlic (*Allium sativum*)**	Non-specified	miRNAs of plant-derived nanovesicles influence microbiota composition	[77]
	Non-specified	Vesicles are efficiently taken up by bone marrow-derived macrophages and inhibit activation of NLRP3 inflammasome	[74]
**Ginger (*Zingiber officinale*)**	Non-specified	Isolated vesicles are taken up by intestinal macrophages; ginger-derived vesicles induce heme oxygenase-1 and IL-10 expression	[76]
	Non-specified	Ginger-derived vesicles protect treated mouse from alcohol-induced liver damage	[80]
	Non-specified	Lipid re-assembled ginger nanovesicles are loaded with doxorubicin; loaded nanovesicles exert apoptotic effects in vitro and reduce tumor growth in vivo	[16]
	Non-specified	Nanovesicles from ginger are efficiently internalized in colon cells after oral administration in treated mice; vesicles prevent and treat chronic colitis and colitis-associated cancer	[47]
	Non-specified	Ginger-derived lipid vehicles loaded with siRNA-CD98 are taken up by colon cells and efficiently decrease CD98 expression in vitro and in vivo	[67]
	Non-specified	Characterization of small RNAs; miRNAs regulate the expression of inflammatory cytokines and cancer-related genes in vitro	[75]
	Non-specified	miRNAs of plant-derived nanovesicles influence microbiota composition	[77]
	Non-specified	Vesicles are efficiently taken up by bone marrow-derived macrophages and inhibit activation of NLRP3 inflammasome	[74]
	Non-specified	Ginger-derived nanovesicles prevent and treat periodontitis in vitro and in vivo	[27]
**Ginseng (*Panax ginseng*)**	Non-specified	Ginseng-derived nanovesicles improved the replicative senescent or senescence-associated pigmented phenotypes of human dermal fibroblasts or ultraviolet B radiation-treated human melanocyte	[81]
**Grape (*Vitis vinifera*)**	Non-specified	Grape exosome-like nanoparticles penetrate intestinal mucus barrier and protect mice from dextran sulfate sodium (DSS)-induced colitis	[17]
	Non-specified	Isolated vesicles are taken up by intestinal macrophages; vesicles induce Nrf2 expression	[76]
	Non-specified	Characterization of grape-derived vesicles	[82]
**Grapefruit (*Citrus paradisi*)**	Non-specified	Grapefruit-derived nanovector inhibit tumor growth in vivo	[42]
	Non-specified	Isolated vesicles are taken up by intestinal macrophages; vesicles induce Nrf2 expression	[76]
	Non-specified	Grapefruit-derived nanovesicles are taken up by intestinal macrophages and ameliorate dextran sulfate sodium (DSS)-induced mouse colitis	[40]
	Non-specified	Intranasally administrated grapefruit nanovesicles deliver miR17 to mice brain tumors	[48]
	Non-specified	Grapefruit-derived lipids carrying miR18 inhibit liver metastasis through induction of M1 macrophages	[37]
	Non-specified	Grapefruit-derived nanovector coated with inflammatory-related receptor enriched membranes of activated leukocytes (IGNVs) are enhanced for homing in on inflammatory tumor tissues	[41]
	Non-specified	Characterization of small RNAs; miRNAs regulate the expression of inflammatory cytokines and cancer-related genes in vitro	[75]
	Non-specified	miRNAs of plant-derived nanovesicles influence microbiota composition	[77]
	Non-specified	Vesicles are efficiently taken up by bone marrow-derived macrophage and inhibit activation of NLRP3 inflammasome	[74]
	Organic agriculture	Nanovesicles from organic agriculture show a higher anti-oxidant level compared to nanovesicles from conventional agriculture	[30]
**Kiwi (*Actinidia chinensis*)**	Non-specified	Characterization of small RNAs; miRNAs regulate the expression of inflammatory cytokines and cancer-related genes in vitro	[75]
	Organic agriculture	Nanovesicles from organic agriculture show a higher anti-oxidant level compared to nanovesicles from conventional agriculture	[30]
**Lavender (*Lavandula*)**	Non-specified	Vesicles are efficiently taken up by bone marrow-derived macrophages and inhibit activation of NLRP3 inflammasome	[74]
**Lemon (*Citrus limon*)**	Non-specified	Lemon-derived nanovesicles inhibits tumor cell proliferation in vitro and tumor growth in vivo	[21]
	Non-specified	Nanovesicles isolated from lemon exert a significant protective effect against oxidative stress	[29]
	Organic agriculture	Nanovesicles from organic agriculture show a higher anti-oxidant level compared to nanovesicles from conventional agriculture	[30]
**Mistletoe (*Viscum album*)**	Non-specified	Vesicles are highly stable and overcome the drying process of plant material	[83]
**Orange (*Citrus sinensis*)**	Non-specified	Characterization of small RNAs; miRNAs regulate the expression of inflammatory cytokines and cancer-related genes in vitro	[75]
**Orange (*Citrus sinensis* ‘Blood Orange’)**	Organic agriculture	Nanovesicles from organic agriculture show a higher anti-oxidant level compared to nanovesicles from conventional agriculture	[72]
**Pear (*Pyrus communis*)**	Non-specified	Characterization of small RNAs; miRNAs regulate the expression of inflammatory cytokines and cancer-related genes in vitro	[75]
**Peas (*Pisum sativum*)**	Non-specified	Characterization of small RNAs; miRNAs regulate the expression of inflammatory cytokines and cancer-related genes in vitro	[75]
**Periwinkle (*Vinca minor*)**	Non-specified	Vesicles are highly stable and overcome the drying process of plant material	[83]
**Soybean (*Glycine soja*)**	Non-specified	Characterization of small RNAs; miRNAs regulate the expression of inflammatory cytokines and cancer-related genes in vitro	[75]
**Strawberry (*Fragaria ananassa*)**	Non-specified	Nanovesicles from strawberry are internalized by human mesenchymal stromal cells and prevent oxidative stress	[84]
**Sunflower (*Helianthus annuus*)**	Non-specified	Characterization of Rab proteins in isolated vesicles	[5]
**Tobacco (*Nicotiana tabacum*)**	Non-specified	Vesicles are highly stable and overcome the drying process of plant material	[83]
**Tomato (*Lycopersicon esculentum*)**	Non-specified	Characterization of small RNAs; miRNAs regulate the expression of inflammatory cytokines and cancer-related genes in vitro	[75]
**Tomato (*Lycopersicon esculentum* ‘Piccadilly’)**	Non-specified	Characterization of an improved isolation method	[85]
**Turmeric (*Curcuma longa*)**	Non-specified	miRNAs of plant-derived nanovesicles influence microbiota composition	[77]
	Non-specified	Vesicles are efficiently taken up by bone marrow-derived macrophages and inhibit activation of NLRP3 inflammasome	[74]
**Watermelon (*Citrullus lanatus*)**	Non-specified	Characterization of watermelon EVs	[86]

## 4. Comments and Discussion

Mammal extracellular vesicles (EV), including exosomes, and have been shown to be involved in an entirely new method of communication between cells; thus, representing a key and rapidly growing research field in science. More recent evidence has shown that plants may also release vesicles of various size (Table 1). Using vegetable- and fruit-derived nanovesicles may well represent a new way to supplement the normal diet of human beings. Through recent technological implementation, nanovesicles can be counted and analyzed in terms of their size and content, which provides a way to standardize the content of the proposed products. A series of recent studies have shown that plant-derived nanovesicles may be taken up by both macrophages and stem cells, with beneficial effects in term of anti-inflammatory and regenerative responses [4,43]. Of course, we need dedicated studies to better understand the reciprocal relationship between plant-derived nanovesicles and human cell-derived nanoparticles, but it is beyond dispute that this represents a new and very promising field, which will open new avenues for human health.

The presence of bioactive molecules in a series of citrus fruits deriving from organic agriculture, including *Citrus limon, Citrus paradisi, Citrus medica, Citrus reticulata, Citrus sinensis, Citrus bergamia, Citrus clementina,* and *Citrus monstruosa,* is excellent news [30]. All the above-mentioned citrus fruits have shown very high levels of a series of anti-oxidants, including ascorbic acid, glutathione, SOD-1, and catalase. Moreover, the anti-oxidant effect from organic agriculture-derived nanovesicles was shown to be significantly higher compared to the same vesicles derived from intensive agriculture. An important achievement in this area was to obtain anti-oxidants without chemical extraction. This provides the anti-oxidants contained in vegetables and fruits in their native and active forms. Moreover, anti-oxidants are protected from immediate oxidation when contained within the lipid membrane of nanovesicles.

All in all, we can claim a series of very important advantages of FVNVs, which can be summarized as follows:Micro/nanovesicles can be obtained from a wide range of fruits and vegetables, and are better if from organic agriculture. Citrus fruits have been by far the more investigated in this area, and we have data that include *Citrus limon*, *Citrus paradisi*, *Citrus medica*, *Citrus reticulata*, *Citrus sinensis* (both yellow and red oranges), *Citrus bergamia*, *Citrus clementina*, and *Citrus monstruosa*.It is possible to perform quality control of any preparation by using standard methods established from the international community in the field (e.g., nanoparticle tracking analysis and immunocapture-based technology).FVNVs contain measurable levels of a series of anti-oxidants, including ascorbic acid (Vit. C), glutathione, superoxide dismutase-1 (SOD-1) and catalase.The anti-oxidants are complexed in a nanostructure that can be called the nano-phyto complex.Due to their structure, FVNVs are stable and contain fully active molecules, showing a clear anti-oxidant effect against both normal and tumor cells.FVNVs may deliver drugs of various origins to target cells, showing significantly higher effects than free molecules.FVNVs can be well used instead of artificial nanoparticles (i.e., liposomes). Compared to the currently available drug delivery systems, plant-derived nanovesicles have multiple advantages, such as low immunogenicity and stability in the gastrointestinal tract (Yang et al., 2018).FVNVs possess high biocompatibility and promises large-scale production.More and more evidence suggests that FVNVs can enter mammalian cells and mediate plant–animal cross-kingdom gene regulation. This evidence show that plant small RNAs packed by plant-derived nanovesicles could survive in their active forms in animals and exogenously modulate the host cellular processes via genetic crosstalk [63]. Moreover, plant-derived nanovesicles have exhibited promising activities in the homeostatic regulation of the immune system, development of tissue engineering and reconstruction, delivery of chemotherapeutic drugs and nucleic acids, etc. [17,43,47,63,66].The above evidence indicates a potential medical application of plant-derived nanovesicles in the regulation of the fundamental biological processes in the human body.

All of these advantages are useful in thinking of a future that can consider vegetable- and fruit-derived nanovesicles for supplementation and drug delivery at a minimum. Of course, we should only consider nanovesicles derived from organic agriculture.

## Figures and Tables

**Figure 1 ijms-23-04919-f001:**
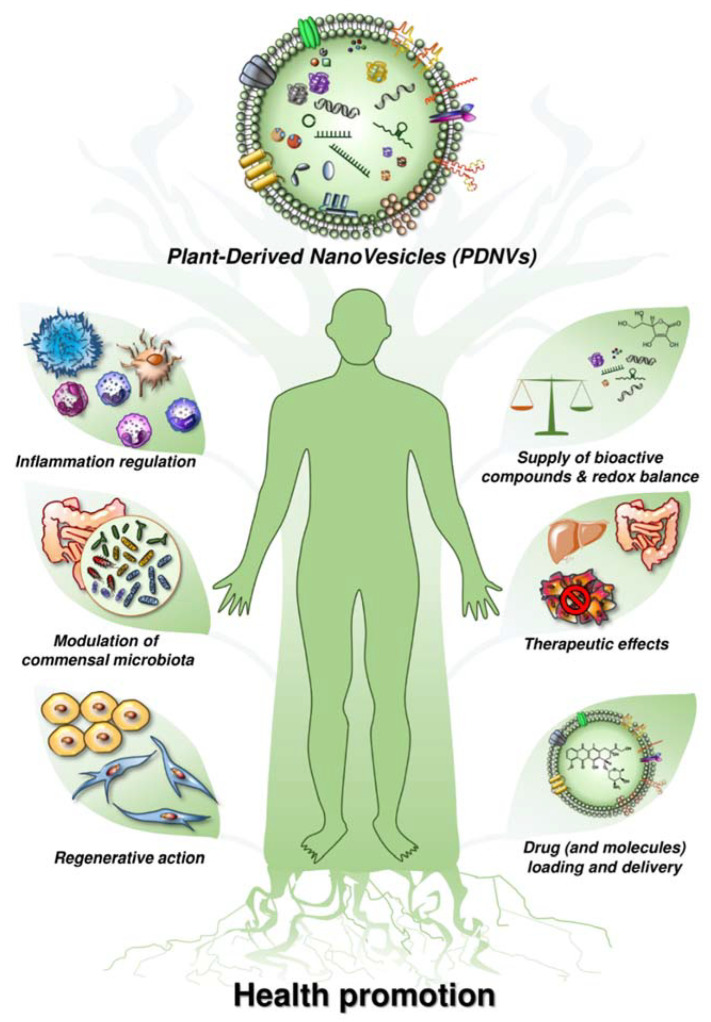
Beneficial effects of plant-derived nanovesicles on human health promotion.

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
