# Peer review of "The Potentiality of Plant-Derived Nanovesicles in Human Health—A Comparison with Human Exosomes and Artificial Nanoparticles"

_ijms, 2022, doi:10.3390/ijms23094919_

Round 1

Reviewer 1 Report

The Reviewer recognizes that the submitted Manuscript “THE POTENTIALITY OF PLANT-DERIVED NANOVESICLES 2 IN HUMAN HEALTH” is devoted to the important and developing rea of knowledge.

However, at present the Reviewer is not able to recommend the MS to be published in the present form.

1) The description is too general.

2) The title is too general, specific medical evidence + clinical trials could be a proof only. Alternatively, the proposed ideas are not more than soft suggestions.

3) Thus, a first claim that we 149 Int. J. Mol. Sci. 2022, 23, x FOR PEER REVIEW 4 of 17

can propose is that nanovesicles extracted from organic agriculture-derived fruits and 150 vegetables may represent “the ideal nanovector for any sort of therapeutic molecule”.

The nanovesicles are consumed with food and then metabolized, any proof of any more serious trials?

4) Figure 1 is absent. Figure 2 is too general.

5) Table 1 is the only serious part of the writing. Unfortunately, it’s a mixture of everything, from a few experiments with cell culture, experiments with healing mice, experiments with plant Arabidopsis and its pathogens, experiments with isolation of transporters from plants vesicles.

6) Unfortunately, the Reviewer is not able the recommend the MS for publication in the present form, the MS is too shallow and general for pretending to be a Review in the good Journal, I am sorry, it could be a descriptive short MS in a low ranking Journal.

Author Response

Point by point reply to reviewers:

WE HAVE CAREFULLY REVISED THE ENGLISH AND THE FORM OF OUR MANUSCRIPT AND A REVISED MANUSCRIPT WITH TRACKED CHANGES ARE ENCLOSED IN THE SUBMISSION

REVIEWER 1

UNFORTUNATELY THIS REVIEWER’REPORT IS CLEARLY A PRIORI HOSTILE, THE REVIEW IS HONESTLY SLOVENLY AND SUGGESTING A CARELESS WAY OF READING; LASTLY IT IS OPENLY IN CONTRAST WITH THE REPORT OF REVIEWER 2. THUS WE ASK THE EDITOR TO NOT CONSIDER IT FOR A FINAL DECISION.

The Reviewer recognizes that the submitted Manuscript “THE POTENTIALITY OF PLANT-DERIVED NANOVESICLES 2 IN HUMAN HEALTH” is devoted to the important and developing rea of knowledge.

However, at present the Reviewer is not able to recommend the MS to be published in the present form.

  • The description is too general.

WE WANTED TO BE GENERAL

2) The title is too general, specific medical evidence + clinical trials could be a proof only. Alternatively, the proposed ideas are not more than soft suggestions.

 IN THE TITLE WE USE “THE POTENTIALITY….”. IN ANY EVENT WE CHANGED THE TITLE ACCORDINGLY BY ADDING “: A COMPARISON TO HUMAN EXOSOMES AND ARTIFICIAL NANOPARTICLES”

3) Thus, a first claim that we 149 Int. J. Mol. Sci. 202223, x FOR PEER REVIEW 4 of 17: ?????? WE CAN’T UNDERSTAND

can propose is that nanovesicles extracted from organic agriculture-derived fruits and 150 vegetables may represent “the ideal nanovector for any sort of therapeutic molecule”.

 ?????????

The nanovesicles are consumed with food and then metabolized, any proof of any more serious trials?

 WE USE TO CURRENTLY EAT THE VAST MAJORITY OF VEGETABLES FRON WHICH NANOVESICLES HAVE BEEN PURIFIED

4) Figure 1 is absent. Figure 2 is too general.

 AMENDED

5) Table 1 is the only serious part of the writing. Unfortunately, it’s a mixture of everything, from a few experiments with cell culture, experiments with healing mice, experiments with plant Arabidopsis and its pathogens, experiments with isolation of transporters from plants vesicles.

 IT IS A SUMMARY OF WHAT WE KNOW ACTUALLY

6) Unfortunately, the Reviewer is not able the recommend the MS for publication in the present form, the MS is too shallow and general for pretending to be a Review in the good Journal, I am sorry, it could be a descriptive short MS in a low ranking Journal.

Reviewer 2 Report

Within the past decade, extracellular vesicles have emerged as a novel cell-cell communication network to regulate various physiological and pathological processes. This is an original review that the authors have drawn up about the Plant-Derived NanoVesicles (PDNV), emphasizing the extracellular vesicles (exosomes). It is important to stress the relationship between plant-derived nanovesicles and human cells-derived nanoparticles reported by authors. They state that it represents, without any doubt also in my point of view, a new and up-and-coming field that will open new avenues for human health. Finally, the manuscript is well-written and clear. After careful consideration, I think this review is appropriate for publication at IJMS.

Author Response

Point by point reply to reviewers:

WE HAVE CAREFULLY REVISED THE ENGLISH AND THE FORM OF OUR MANUSCRIPT AND A REVISED MANUSCRIPT WITH TRACKED CHANGES ARE ENCLOSED IN THE SUBMISSION

REVIEWER 2

Within the past decade, extracellular vesicles have emerged as a novel cell-cell communication network to regulate various physiological and pathological processes. This is an original review that the authors have drawn up about the Plant-Derived NanoVesicles (PDNV), emphasizing the extracellular vesicles (exosomes). It is important to stress the relationship between plant-derived nanovesicles and human cells-derived nanoparticles reported by authors. They state that it represents, without any doubt also in my point of view, a new and up-and-coming field that will open new avenues for human health. Finally, the manuscript is well-written and clear. After careful consideration, I think this review is appropriate for publication at IJMS.

WE THANK THE REVIEWER FOR HIS/HER EXAUSTIVE AND FAIR COMMENTS
